# The Effect of Fabry Disease Therapy on Bone Mineral Density

**DOI:** 10.3390/diseases12050102

**Published:** 2024-05-13

**Authors:** Tess Aitken, Mark K. Tiong, Andrew S. Talbot, Irene Ruderman, Kathleen M. Nicholls

**Affiliations:** 1Department of Nephrology, The Royal Melbourne Hospital, Parkville 3050, Australia; 2Department of Medicine (RMH), The University of Melbourne, Parkville 3010, Australia

**Keywords:** Fabry disease, lysosomal storage diseases, bone mineral density, enzyme replacement therapy, chaperone therapy

## Abstract

Fabry disease (FD) is an X-linked lysosomal storage disorder, characterised by the cellular accumulation of globotriaosylceramide due to impaired alpha-galactosidase A enzyme activity. FD may manifest with multisystem pathology, including reduced bone mineral density (BMD). Registry data suggest that the introduction of Fabry-specific therapies (enzyme replacement therapy or chaperone therapy) has led to significant improvements in overall patient outcomes; however, there are limited data on the impact on bone density. The aim of this study was to describe the effect of Fabry-specific therapies on longitudinal changes in bone mineral density (BMD) in FD. We performed a retrospective observational study analysing bone densitometry (DXA) in patients with genetically confirmed FD. Patients were grouped based on the use of Fabry-specific therapies. The between-group longitudinal change in BMD Z-score was analysed using linear mixed effects models. A total of 88 FD patients were analysed (50 untreated; 38 treated). The mean age at first DXA was 38.5 years in the untreated group (84% female) and 43.7 years in the treated group (34% female). There was no significant longitudinal between-group difference in the BMD Z-score at the lumbar spine. However, the Z-score per year at the total hip (β = −0.105, *p* < 0.001) and femoral neck (β = −0.081, *p* = 0.001) was significantly lower over time in the treated than the untreated group. This may reflect those receiving therapy having a more severe underlying disease. Nevertheless, this suggests that Fabry-specific therapies do not reverse all disease mechanisms and that the additional management of BMD may be required in this patient population.

## 1. Introduction

Fabry disease (FD; OMIM 301500) is a genetic X-linked lysosomal storage disorder caused by an error in the glycosphingolipid pathway due to deficiencies in the lysosomal enzyme α-galactosidase A (α-Gal A; EC 3.2.1.22) [1]. Reduced or absent α-Gal A activity results in the widespread intracellular accumulation of globotriaosylceramide (Gb3). This triggers cellular damage across multiple organ systems, with cardiovascular, cerebrovascular and renal disease considered to be the most life-threatening manifestations [2]. However, FD may present with a wide range of phenotypes, stemming from differences in mutations and residual α-Gal A activity. Further, despite being an X-linked condition, heterozygous females may still present with severe disease complications based on variations in X inactivation [3]. The exact mechanisms linking Gb3 accumulation to progressive organ damage are unclear; however, cellular hypertrophic responses and the induction of inflammatory and fibrotic pathways have all been implicated [1].

Patients affected by FD have been found to have lower bone mineral density (BMD), as evidenced by higher incidences of osteopenia and osteoporosis, particularly in males [4,5,6]. The cause of bone disease in FD patients is likely to be multifactorial. Malnutrition, a low body mass, a low level of outdoor activity, the malabsorption of calcium and vitamin D and chronic kidney disease are all prevalent in FD and associated with reduced BMD. Further, neuropathic pain is a classical feature of FD and often results in treatment with antiepileptic drugs (AED), which are also thought to predispose patients to bone loss [5].

Two main treatment modalities are now available for the treatment of FD. Both aim to partially restore α-Gal A activity. Enzyme replacement therapy (ERT) involves regular intravenous infusions of a recombinant enzyme (agalsidase alpha or agalsidase beta). ERT has been shown to reduce the plasma and tissue concentrations of Gb3 and improve the clinical symptoms of FD [7,8]. Oral chaperone therapy (migalastat) binds to misfolded alpha-Gal A enzymes, and, in individuals with “amenable” mutations, it stabilises the immature enzyme for transport to the lysosome, resulting in increased enzyme activity [9]. Despite evidence of treatment efficacy on cardiovascular effects, pain, neurological and gastrointestinal symptoms, there is minimal information about whether ERT and chaperone therapy alter bone outcomes. In Gaucher disease, another lysosomal storage disorder, Sims et al. reported that the commencement of ERT was associated with a significant increase in bone density [10]. In comparison, limited comparable analysis exists in FD, where a bone phenotype is less pronounced. Recently, Nose et al. found that the commencement of ERT was associated with increases in BMD over two years among males with FD [11]. However, this was a small cohort study and longitudinal BMD data were only available for ten individuals with FD.

We aimed to describe whether there are any differences in longitudinal trends in bone density between those being treated with Fabry-specific therapies and those who are not.

## 2. Materials and Methods

We performed a retrospective observational study to describe the bone density patterns in a cohort of individuals with FD being managed by a tertiary referral centre. This study was approved by the local ethics committee (Melbourne Health Human Research Ethics Committee, reference number MH 2015.029).

### 2.1. Participants, Treatment Groups and Baseline Definition

We included all adult patients under the care of a single quaternary referral centre who had a genetically confirmed diagnosis of FD and at least one measurement of bone density by dual-energy X-ray absorptiometry (DXA) between 1 January 2001 and 31 December 2022. Patients who did not have at least one measurement of bone density were excluded. We performed a retrospective review of the electronic medical records, pathology and DXA examinations.

Patients were grouped into “treated” and “untreated” based on whether they had received treatment with a Fabry-specific therapy during the study period. Participants were included in the treated group if they had received ERT (regular intravenous infusions with agalsidase alpha or agalsidase beta) or chaperone therapy (orally administered migalastat). We included all patients who had received one of these therapies, including those who had subsequently transitioned to an alternative Fabry-specific therapy.

We followed patients from the study baseline date until the time of death or the conclusion of the study. For the untreated group, the baseline was taken as the date of the first DXA. For the treated group, the baseline date was defined as either the date of the first DXA after the commencement of Fabry-specific therapy, or the date of an available DXA taken up to one year before commencement (whichever occurred first). We had initially planned to include all scans performed before the commencement of Fabry-specific therapy in the treated group to analyse changes in bone density before and after the commencement of treatment. However, only two individuals in the treated group had scans available (4 scans in total) more than one year before the commencement of therapy.

### 2.2. Assessment of Bone Density

At our institution, bone densitometry is performed by DXA at initial review and then every two years as part of the standard of care for our Fabry disease service. However, the exact timing and frequency of these exams is variable and is influenced by patient preferences and logistical considerations (e.g., patients living a long distance from our centre and unable to readily attend scanning appointments). Bone density was assessed by our institution’s accredited bone densitometry unit using a Hologic Horizon QDR-4500A (prior to 2018) or a Hologic Horizon A (from 2018) machine, with appropriate calibration and quality control measures conducted during the changeover of the machines. All scans were performed by a qualified technician and the machine was calibrated daily using a phantom and according to the manufacturer’s recommendations. Bone density (g/cm^2^) was measured at the lumbar spine, total hip and femoral neck, and T- and Z-scores were calculated based on the third National Health and Nutrition Examination Survey reference population [12].

### 2.3. Other Variables

Participant demographics, kidney function (estimated glomerular filtration rate using the Chronic Kidney Disease Epidemiology Collaboration equation) and other routine biochemistry parameters were recorded at the time of baseline DXA. In addition, we also recorded medications at baseline, including the use of AED, given that they have been associated with an increased risk of osteoporosis and are commonly used for the therapeutic management of neuropathic pain in Fabry disease [5].

### 2.4. Statistical Analysis

Demographic, standard biochemical and baseline medication and DXA data are presented using descriptive statistics. For continuous variables, the mean and standard deviation are presented or the median and interquartile range are presented for normally distributed and skewed data, respectively. Numbers and percentages are presented for categorical variables. Baseline differences between groups were tested using a *t*-test or Kruskal–Wallis test for normally and skewed continuous variables, respectively, and a chi-square test for categorical variables.

In order to examine whether Fabry-specific treatment is associated with any differences in the longitudinal change in bone density, we fitted linear mixed effects models with repeated measures. This approach accounts for intra-individual correlations between repeated observations while also being robust to a variable number of observations between individuals. Because our cohort was heterogenous and bone density varies with age and sex, we utilised Z-scores for the longitudinal assessment of bone density, as suggested by Sims et al. [10], given that this measure is normalised for the population age and sex. This also allowed for the examination of changes in bone density over time, while accounting for sex-specific, age-expected (non-linear) changes. We fitted separate models for the Z-scores at the lumbar spine, total hip and femoral neck. For our main models, the treatment group, time and group–time interaction were modelled as fixed effects. We included a random intercept for each participant to allow for individual differences and correlations between repeated measures, and we used a restricted maximum likelihood approach with an unstructured covariance structure. The normal distribution of residuals was confirmed after fitting each model.

### 2.5. Sensitivity and Post Hoc Exploratory Analyses

As Z-scores normalise bone density to an age- and sex-adjusted reference population, in our main analyses, we did not adjust for baseline differences in these demographic variables (Model 1). However, as part of a sensitivity analysis, we refitted each model with adjustment for age and sex (Model 2). In addition to age and sex, we also fitted models with further adjustment for baseline body mass index (BMI), kidney function (estimated glomerular filtration rate), AED use and smoking history (Model 3).

Patients with Fabry disease are at an increased risk of chronic kidney disease (CKD) and kidney-related bone disease. This risk increases as the kidney function declines and is particularly high in those with kidney failure requiring dialysis or recipients of a kidney transplant [13,14]. As such, we also performed sensitivity analyses, where we excluded the DXA data taken after a patient commenced dialysis or received a kidney transplant.

We also fitted models excluding individuals who had a significant interruption in their treatment (defined as being on treatment for less than 80% of the participant’s accrued study observation time).

In our main linear mixed effects models, there appeared to be a significant between-group difference for the total hip and femoral neck Z-scores. In order to further examine whether this represented a longitudinal change from the baseline within either group, in a post hoc exploratory analysis, we fitted separate linear mixed effects models with data restricted to either treated or untreated individuals for these two parameters.

Two-tailed *p* values < 0.05 were considered significant. All data were analysed using Stata version 17.0 (StataCorp, College Station, TX, USA) and figures were produced using GraphPad Prism version 9.0.0 (GraphPad Software, San Diego, CA, USA).

## 3. Results

A total of 88 individuals met the inclusion criteria and were included in this analysis. Within this cohort, 38 patients had received treatment with at least one Fabry-specific therapy and were included in the treated group, whereas 50 had not. The baseline demographics of the cohort are presented in Table 1 and Appendix A. The age at baseline was comparable between groups (38.5 years for the untreated group vs. 43.7 years for the treated group, *p* = 0.087); however, consistent with the X-linked inheritance pattern of Fabry disease, the proportion of female participants was lower in the treated group than the untreated group (34% vs. 84%, *p* < 0.001). Similarly, as expected, individuals in the treated group had lower eGFR at baseline (84 vs. 102 mL/min/1.73 m^2^, *p* = 0.005) and were more likely to be using anti-epileptic medications for neuropathic pain (51% vs. 18% *p* = 0.001).

Among the 38 participants in the treated group, 20 participants initially received treatment with agalsidase alpha, while 17 initially received agalsidase beta and one migalastat (Appendix A). The mean age at the commencement of treatment was 39.1 ± 13.4 years, and the median time between the commencement of treatment and the baseline DXA was 1.4 [0.4–2.4] years.

### 3.1. Baseline and Longitudinal Change in Bone Density between Groups

The DXA values for bone density, T-scores and Z-scores for the cohort at baseline are displayed in Table 1. The bone mineral density, T-scores and Z-scores at baseline were comparable between the groups at the lumbar spine, total hip and femoral neck.

During the follow-up period, individuals in the untreated group had a median of three [IQR 1–3] DXA scans over a median of 4.9 [range 0–12.3 years] years of follow-up time. By comparison, individuals in the treated group had a median of 3.5 [IQR 3–6] DXA scans over a median of 12.8 [range 0.2–21.7] years.

The Z-scores of the lumbar spine, total hip and femoral neck for the untreated and treated groups are plotted in Figure 1, and linear mixed effects model coefficients are presented in Table 2.

For the lumbar spine, there did not appear to be a significant between-group difference in the longitudinal change in the Z-score (β = −0.026 [95% CI −0.073, 0.021], *p* = 0.278). In comparison, the longitudinal change in the total hip Z-score was significantly lower in the treated group than in the untreated group (β = −0.081 [95% CI −0.105, −0.060], *p* < 0.001). Similarly, the longitudinal change in the femoral neck Z-score was also lower for the treated group (β = −0.081 [95% CI −0.128, −0.033], *p* = 0.001).

### 3.2. Sensitivity Analyses

The linear mixed effects models for the lumbar spine, total hip and femoral neck Z-scores were refitted with sequential adjustment for the baseline covariates (Table 2). The adjustment for age and sex (Model 2) and the additional adjustment for BMI, kidney function, AED use and smoking history (Model 3) did not significantly alter the direction of magnitude of the main, unadjusted models.

The exclusion of bone density data obtained after patients commenced dialysis or received a kidney transplant (six individuals, all in the treated group) did not substantially change the magnitude or significance of the between-group difference in Z-score for the lumbar spine, total hip or femoral neck in the unadjusted and multivariate adjusted models. Similarly, the exclusion of two individuals in the treated group who had significant interruptions to their treatment (receiving treatment for less than 80% of their accrued follow-up time from baseline) did not alter our main results.

### 3.3. Post Hoc Exploratory Analysis

Given the finding that individuals in the treated group appeared to have significantly lower total hip and femoral neck Z-scores over time, the linear mixed effects models were refitted after restricting the data to each group (Appendix A). The mean Z-scores in the untreated group appeared to increase over time, while the Z-scores in the treated group did not show any significant longitudinal change.

## 4. Discussion

In this retrospective observational cohort study, we found that FD patients who were receiving treatment with Fabry-specific therapies had worse bone density trajectories in comparison to those with FD who were not receiving Fabry therapies. Specifically, the Z-scores for the total hip and femoral neck were significantly higher in the untreated group in comparison to the treated group.

The reduction in bone mineral density is a well-documented component of the multi-system effects of Fabry disease [4,5,6]. The introduction of Fabry-specific therapies, including enzyme replacement therapies with agalsidase alpha and agalsidase beta, and, more recently, the chaperone migalastat, have substantially improved the prognosis of Fabry disease over the last two decades [15,16]. However, there are limited previous data describing the effects of Fabry-specific therapies on bone outcomes [11].

This study suggests that despite receiving treatment with Fabry-specific therapies, bone density is significantly worse among those receiving treatment compared to those who are not. It is plausible that the difference between groups is attributable to selection bias, given that those receiving treatment are more likely to have severe manifestations of FD. FD is a heterogenous condition, with a wide spectrum of phenotypes and disease severity [2]. In Australia, in order to receive reimbursed treatment with Fabry-specific therapy, individuals must have confirmed evidence of Fabry-related kidney disease, heart disease, stroke or significant neuropathic pain [17]. Because of this, individuals in the treated group would, in general, be expected to have a more severe underlying disease than those in the untreated group. Consistent with this, the treated group had a lower proportion of females, lower baseline eGFR and a higher proportion of individuals who were receiving regular antiepileptic medication for neuropathic pain. Further, in a cohort of 15 individuals with FD, Nose et al. recently reported that bone density was inversely correlated with plasma globotriaosylsphingosine (a biomarker in FD) [11]. Unfortunately, we did not have Fabry-specific biomarkers universally available for our cohort and so were not able to utilise these to attempt to further quantify the disease severity in our study. Nevertheless, we acknowledge that these biomarkers also have limitations, particularly when measured in female patients and in those with “non-classical” genetic mutations [18].

The exact pathobiology underlying poor bone outcomes in FD remains incompletely understood; however, poor nutrition, chronic kidney disease [6], chronic inflammation [19], endocrine dysfunction [20], reduced weight-bearing exercise due to physical limitations and the use of antiepileptic medications for neuropathic pain [5] are all features of FD that could plausibly contribute. It is anticipated that many of these factors would be more severe in individuals with FD warranting Fabry-specific therapy. The between-group difference in the longitudinal change in hip and femoral neck bone density persisted after adjustment for differences in participant demographics (Model 2) and for the baseline BMI, eGFR, AED use and smoking history (Model 3). However, we were unable to interrogate the relationship between bone density and any additional potentially contributing factors, given that these were not routinely collected in our cohort.

Our results add further context to the findings of Nose et al., who observed increases in bone density after the commencement of ERT in a small group of individuals with FD (five patients, all male) [11]. In this study, the increase in bone density was seen after two years of ERT. In contrast, the median time between the commencement of Fabry-specific therapy and baseline DXA in our study was 1.4 years, and this may explain why we did not detect a similar early increase in bone density in our treated group. Nevertheless, we provide longer-term observational data from a larger cohort of individuals with FD, including those on and not on Fabry-specific therapies. Our findings should be considered complimentary rather than contradictory to those of Nose et al. Even if some individuals with FD experience an initial increase in bone density after the commencement of Fabry-specific therapy (as described by Nose et al.), our data suggest that, in the long term, the their bone density trajectories will still be significantly lower than their peers with FD who are not on treatment (and who likely have less severe underlying FD). While the introduction of Fabry-specific therapies has led to improved overall outcomes, it is apparent that the treatment may slow, but not completely halt, the progression of other significant events, such as stroke, a decline in eGFR and cardiac events [21,22,23]. Therefore, one explanation for our findings, and the findings of Nose et al., is that treatment with Fabry-specific therapies may improve but not completely reverse the underlying causes of adverse bone metabolism in FD. This would imply that individuals with FD who are receiving treatment with Fabry-specific therapies still require additional measures to assess and manage their bone health.

Of interest, in our exploratory post hoc analysis, where individual groups were separately examined, it appeared that the total hip and femoral neck Z-scores increased over time in the untreated group, whereas no such change was demonstrated in the treated group. This implies that the observed between-group difference in these parameters in the main analysis was primarily driven by an increase in the Z-score in the untreated group, rather than a decrease in the Z-score in the treated group. We utilised Z-scores in this study given the small numbers of our cohort and the diverse age profile and uneven sex composition. While a progressive increase in Z-score ostensibly implies an abnormal increase in bone density in the untreated group, it is important to acknowledge the limitations of using Z-scores in this longitudinal manner. Bone density Z-scores normalise an individual’s bone density to an age and sex (and sometimes ethnically) matched reference population, and they are recommended by international authorities for the enhanced detection of bone density below the “expected range for age” [24]. However, it should not necessarily be inferred that individuals are expected to follow the same Z-score curve over time. This is because the reference population data are ethnicity-specific and are predominantly derived from large cross-sectional studies [12,25,26]. The latter point has been illustrated by previous longitudinal studies that followed individuals with serial bone density scans, rather than the single measures of bone density that are typically used to generate reference population data. In several of these cohort studies, the rate of observed loss in bone density with age was found to be lower than the rate that would have been predicted from cross-sectional data [27,28,29]. Whilst unclear, it has been suggested that this discrepancy between cross-sectional and longitudinal data may be related to differences in the birth cohort, which are not accounted for in cross-sectional studies [29]. Consistent with this, recent evidence suggests that, globally, the age-adjusted incidence of osteoporosis in the general population may be decreasing over time [30,31].

Our use of longitudinal data, as well as a non-ethnically matched reference population, provide plausible explanations for the observed increase in the Z-scores in the untreated group. We used reference population Z-scores to provide a framework to perform between-group comparisons of bone density in our heterogenous cohort, given that age-related changes in bone mass follow a non-linear and sex-specific trajectory [29,32]. Given our small numbers, we did not directly model changes in absolute bone density as a sex-stratified function of age for our primary analysis. In the post hoc exploratory analysis, we did, however, note that the between-group differences seen in the overall cohort appeared to be broadly consistent when the analyses were repeated separately for female and male patients (Appendix A). However, considering the limited numbers of patients in each of these sub-analyses, particularly for males (n = 33), future studies of larger cohorts would be valuable to more thoroughly interrogate any sex-specific differences in bone outcomes in this population. Whilst acknowledging the limitations of Z-scores, our primary interest in this study was to compare differences in the longitudinal change in bone density between those being treated and those not being treated with Fabry-specific therapies. Given that both groups were subjected to BMD testing in the same facility, and that the results were standardised against the same reference population to derive the Z-scores, it is still notable that there was a clear longitudinal differentiation in the Z-scores between groups, even after adjustment for demographic and other available potential confounders.

### Limitations

We acknowledge that this study has a number of important limitations. This is a small, retrospective observational study. As already discussed, the small numbers limited our ability to perform more in-depth modelling of the longitudinal changes in bone density. Similarly, it would require much larger numbers to examine patient-level outcomes, such as bone fractures, instead of bone density measured non-invasively by DXA, as we have done here. We had also initially planned to examine the trajectory of the change in bone density before and after the commencement of Fabry-specific therapy; however, only two individuals had multiple bone density scans performed before and after the commencement of therapy. As such, we relied on comparing those receiving Fabry-specific therapy to those who were not, and we acknowledge that the individuals were likely to be inherently different between groups. Nevertheless, FD is a rare condition, and, given that enzyme replacement therapy and chaperone therapy have become established therapies, it is unlikely that an interventional controlled trial examining the effects of Fabry-specific therapies on bone outcomes will be feasible in the future [16]. Bone turnover markers were not widely available for this cohort; however, these could be helpful in future studies to further understand the impact of Fabry-specific therapies on bone metabolism, particularly if collected before and after the commencement of treatment. Finally, only one patient in the treated group received migalastat, while the remaining 37 individuals in the treated group received enzyme replacement therapy. This precluded us from investigating any differences between treatment modalities. Further study of the bone density in patient cohorts enriched in individuals being treated with migalastat may be of particular interest in the future.

## 5. Conclusions

In this study, we found that the total hip and femoral neck bone density was significantly worse over time in those with FD who were receiving Fabry-specific therapies in comparison to those who were not. This persisted after adjustment for baseline differences in demographics, kidney function, BMI and the use of AED; however, differences in the underlying severity of FD are likely and could not be adjusted for. Nevertheless, our results imply that the current FD therapies may be insufficient to halt, or reverse, the effects of FD on bone density. As a result, it may be prudent to consider additional treatments specifically targeting bone health in this patient population.

## Figures and Tables

**Figure 1 diseases-12-00102-f001:**
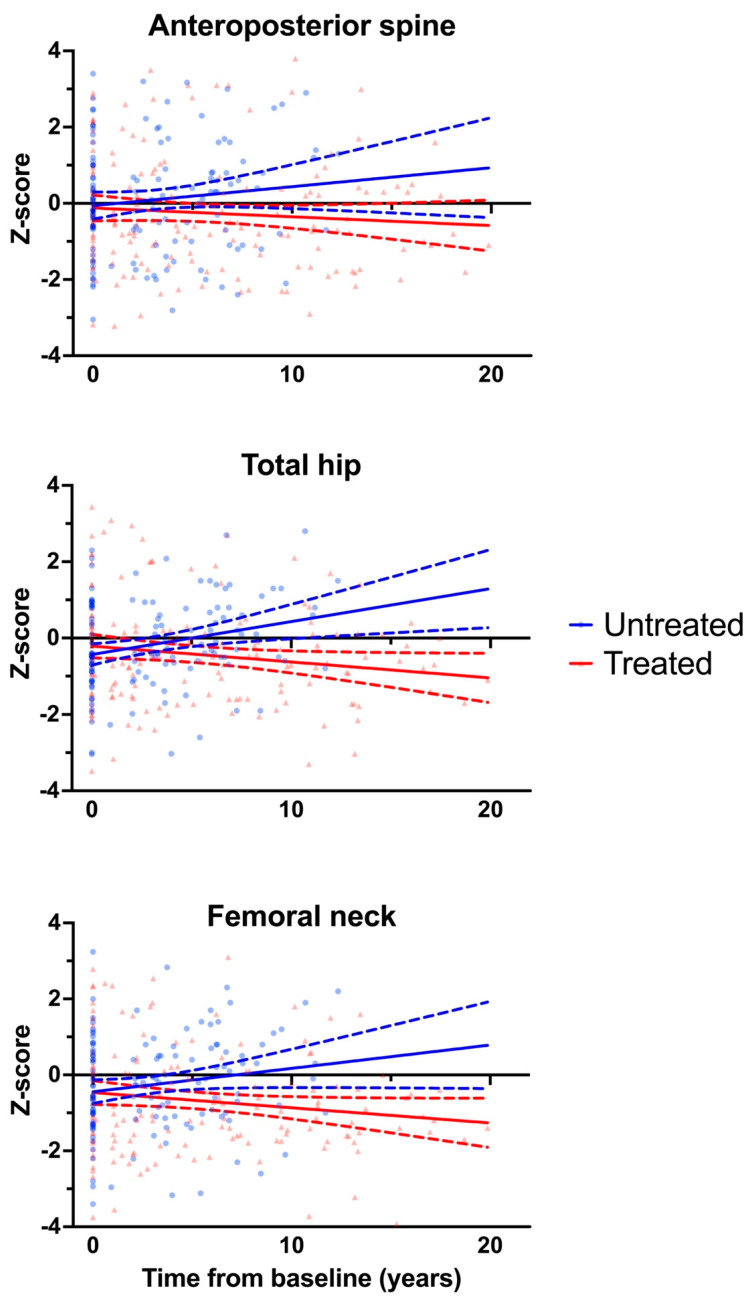
Longitudinal change in bone density Z-score by group. Serial measurements of bone mineral density (Z-score) by group. Fitted linear regression line with 95% confidence interval (dashed lines) shown (by group).

**Table 1 diseases-12-00102-t001:** Baseline demographics.

	Overall (n = 88)	Untreated (n = 50)	Treated (n = 38)	*p*-Value for Between-Group Difference
Baseline demographics				
Age—yrs	40.8 ± 14.1	38.5 ± 15.5	43.7 ± 11.7	0.087
Female sex	55 (62.5)	42 (84.0)	13 (34.2)	<0.001
Body mass index—kg/m^2^	24.2 [21.7–27.9]	24.2 [22.2–30.1]	24.2 [20.9–27.4]	0.293
eGFR—mL/min/1.73 m^2 #^	94.7 ± 28.8	101.8 ± 21.2	84.2 ± 34.9	0.005
Serum calcium (adjusted)—mmol/L ^##^	2.37 ± 0.12	2.38 ± 0.12	2.35 ± 0.11	0.303
Serum phosphate—mmol/L	1.10 ± 0.20	1.10 ± 0.17	1.10 ± 0.23	0.978
Smoking history (current or previous)	24 (27.3)	12 (24.0)	12 (31.6)	0.429
Anti-epileptic medication use ^###^	28 (32.2)	9 (18.0)	19 (51.4)	0.001
Bone mineral density				
*Lumbar spine*				
BMD—g/cm^2^	1.02 ± 0.13	1.01 ± 0.14	1.04 ± 0.13	0.257
T-score	−0.34 ± 1.26	−0.36 ± 1.28	−0.32 ± 1.25	0.874
Z-score	0.06 ± 1.43	0.06 ± 1.43	0.07 ± 1.46	0.956
*Total hip*				
BMD—g/cm^2^	0.92 ± 0.15	0.90 ± 0.13	0.95 ± 0.17	0.150
T-score	−0.65 ± 1.33	−0.62 ± 1.21	−0.69 ± 1.49	0.808
Z-score	−0.26 ± 1.34	−0.33 ± 1.21	−0.18 ± 1.50	0.600
*Femoral neck*				
BMD—g/cm^2^	0.80 ± 0.14	0.80 ± 0.13	0.81 ± 0.14	0.814
T-score	−1.07 ± 1.45	−0.92 ± 1.37	−1.3 ± 1.54	0.265
Z-score	−0.39 ± 1.43	−0.37 ± 1.31	−0.41 ± 1.59	0.902

Mean ± standard deviation, median [interquartile range] or number (%); *p*-value for baseline differences between groups were tested using *t*-test or Kruskal–Wallis test for normally and skewed continuous variables, respectively, and chi-square test for categorical variables. eGFR calculated according to the Chronic Kidney Disease Epidemiology Collaboration equation. For the untreated group, the baseline was taken as the date of the first DXA. For the treated group, the baseline date was defined as either the date of the first DXA after the commencement of Fabry-specific therapy, or the date of an available DXA taken up to one year before commencement (whichever occurred first). ^#^ eGFR data were missing for four individuals in the treated group (including one on haemodialysis at baseline). ^##^ Calcium and phosphate data were missing for six individuals (all in the treated group). ^###^ Medication data were missing for one individual in the treated group.

**Table 2 diseases-12-00102-t002:** Longitudinal change in bone density by treatment.

	Group by Time Interaction from Linear Mixed Effects Model		
	β-Coefficient	95% Confidence Interval	*p*-Value
Lumbar spine Z-score			
Model 1	−0.026	−0.073, 0.021	0.279
Model 2	−0.025	−0.072, 0.023	0.311
Model 3	−0.026	−0.078, 0.027	0.334
Total hip Z-score			
Model 1	−0.105	−0.150, −0.060	<0.001
Model 2	−0.105	−0.151, −0.060	<0.001
Model 3	−0.110	−0.158, −0.061	<0.001
Femoral neck Z-score			
Model 1	−0.081	−0.128, −0.033	0.001
Model 2	−0.081	−0.129, −0.032	0.001
Model 3	−0.090	−0.142, −0.038	0.001

Beta coefficient is for group-by-time change in Z-score. Model 1 = unadjusted. Model 2 = adjusted for baseline age and sex. Model 3 = Model 2, plus additional adjustment for baseline body mass index, eGFR, antiepileptic medication use, smoking history.

## Data Availability

The data that support the findings of this study may be made available on request to the corresponding author. The data are not publicly available due to privacy reasons.

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
