# Peer review of "The Effect of Fabry Disease Therapy on Bone Mineral Density"

_diseases, 2024, doi:10.3390/diseases12050102_

Round 1

Reviewer 1 Report

Comments and Suggestions for Authors

This study examined longitudinal Fabry patient registry data to examine the effect of current treatments on bone density, which is reduced with progression of Fabry disease. The surprising conclusion of this study is that bone density in treated Fabry patients is less than in untreated Fabry patients. The authors do a good job of examining this data in different ways and interpreting the data without over-interpretation. They also well-describe the limitations of this data. Overall, their results suggest that further examination of bone parameters in Fabry patients is warranted to determine to what extent current treatments (ERT or chaperone therapy) can alter bone phenotypes.

Two issues that must be addressed are related to the description of the patient data:

1) Define eGFR.

2) Describe whether the biochemical endpoints (eGFR, Ca, Ph) examined were measured in urine or serum.

Reviewer 2 Report

Comments and Suggestions for Authors

This paper investigates the impact of therapy on Bone Mineral Density (BMD) in patients with Fabry disease (FD). It is common for patients with lysosomal diseases to experience a decrease in BMD. While many studies have explored the effects of decreasing BMD and the development of early-age osteoporosis in Gaucher disease, there is a lack of research on the natural history of bone remodeling and the impact of enzyme replacement therapies (ERT) or substrate reduction therapy (SRT) on BMD in patients with Fabry disease.

I would like to bring to attention that FD is a disease that is inherited in an X-linked diagonal manner. This means that sex differences play a significant role in the pathology of FD. Males with a single X chromosome are highly affected by the disease, while heterozygous females are affected later with different symptomatic dynamics. Moreover, therapy management differs between male and female patients, including when to initiate and what type of therapies to use.

Additionally, It is important to note that reduced bone mineral density (BMD) and the development of osteopenia or osteoporosis are influenced by gender. Women are at a higher risk of developing these conditions due to their smaller and less dense bones compared to men, as well as the regulation of bone remodeling by sex hormones and age. This risk further increases during menopause when estrogen levels drop, which is a key hormone for maintaining bone remodeling.

However, this paper investigates BMD in male and female FD patients together. I am not sure “why?” As a reviewer, I feel that this paper has the potential to be an interesting piece of research. However, a lot of additional work is needed to move beyond the "statistical numeral values without real data support" and the conclusions drawn in this paper.

Major comment:

-          Please demonstrate analyses are separated by gender, distinguishing between males and females in tables and figures.

-          Please ensure that the analysis is separated by age for females before and after 45. Also, compare the correlation between BMD and age in FD vs. the general population, providing proof that decreasing BMD is related to Fabry disease.

-          The authors only provided a summary of their final statistical analysis, which included a brief explanation of how they normalized BMD to an age and sex-adjusted reference population. However, the authors did not provide any actual data to support their findings. It is unclear why the authors chose not to present the real data, as it makes it difficult for readers to evaluate the accuracy of the results. Without the actual data, it is challenging to understand how the authors reached their conclusions.

-          The study demonstrated that the Z-score in the hip and femoral neck was higher in the untreated group than in the treated group. Please present the actual data to support this conclusion. Graphs showing individual patients' Z or T scores in relation to males/females, age, ERT, and SRT will help to understand the results.

-          Figure 1. Longitudinal change in bone density Z-score by group. What do all these dots mean? If these individual patients, it is unclear how lines were created. There is a 20-year difference between the beginning and end of the data. If the patient is female and was 35 years old at the start of the analysis and 55 years old at the end of the data, it's possible that decreasing BMD may not be related to FD? A lot of data is missing. Normal BMD -1 to up, not from zero

-          A detailed table with patient cohorts, including ERT and SRT treatments, males/females, and other manifestations related to Fabry disease, as a cardio and/or kidney, was missing. Classic cardio vs. nonclassic cardio. 

Reviewer 3 Report

Comments and Suggestions for Authors

The authors conducted a retrospective observational study analysing longitudinal changes in BMD among FD patients treated with enzyme replacement therapy or chaperone therapy compared to untreated patients. The article presents valuable insights into the relationship between Fabry-specific therapies and bone density outcomes. Overall, the language and presentation of the manuscript are appropriate. Addressing the few points would further enhance the quality of the article.

 1. The authors should briefly outline the exclusion criteria for patient selection as it currently lacks clarity.

 2. Since only one patient was included for Oral chaperone therapy (migalastat), the association of this therapy with BMD may not be logically supported. This limitation should be explicitly acknowledged in the study.

 3. Discussion on potential mechanisms underlying the observed differences in bone density outcomes between treated and untreated groups would add valuable insights to the discussion.

4. As per the topic of the manuscript, consider enlarging and adding more details to the introduction/discussion part with referencing the following article to nicely complete the paper.

"Izhar R, Borriello M, La Russa A, Di Paola R, De A, Capasso G, Ingrosso D, Perna AF, Simeoni M. Fabry Disease in Women: Genetic Basis, Available Biomarkers, and Clinical Manifestations. Genes (Basel). 2023 Dec 26;15(1):37. doi: 10.3390/genes15010037".

Comments on the Quality of English Language

Minor revision required.

Reviewer 4 Report

Comments and Suggestions for Authors The authors aimed  to describe whether there are any differences in longitudinal trends in bone density between those being treated with Fabry specific therapies, and those who are not. The introduction is well written , with adequate bibliographic references . The objective of the study is   established to matherial and methods and must be changed

The methodology is complete, widely described, which would allow the study to be carried out by another research group. However  the study has the limitations of the retrospective evaluation

The results are clear expressed  and easy to understand The discussion is adapted to the results obtained.

Round 2

Reviewer 2 Report

Comments and Suggestions for Authors

I disagree with the author's conclusion that analyzing males and females separately is not sufficient in the case of a cohort of 88 patients with Fabry disease, where 55 patients are females. In my view, it is sufficient to analyze them separately.

Additionally, it is important to analyze males and females separately for two reasons. First, males and females with Fabry disease exhibit different levels of disease manifestation due to it being an X-linked lysosomal disorder. Second, diseases related to bone mineral density are gender-specific. For instance, at age 50, one in two women is affected by osteoporosis, while only one in five men is affected by it. 

Round 3

Reviewer 2 Report

Comments and Suggestions for Authors

·         Please consider adding a clear explanation of the statistical analysis for Supplemental Table 4. While post-hoc analysis is appropriate for large cohort numbers, another statistical design could be utilized for smaller datasets, depending on the question. Clarification in the text could improve the understanding of your research. 

·         The authors have mentioned in the material and methods chapter “Statistical analysis”, that they plan to examine changes in bone density while accounting for sex- and age-specific changes. 

“Because our cohort is heterogenous and bone density varies with age and sex, we utilized Z-scores for longitudinal assessment of bone density as suggested by Sims et al [10], given that this measure is normalized for population age and sex. This also allows for the examination of changes in bone density over time while accounting for sex-specific, age-expected (non-linear) changes. We fitted separate models for Z-scores at the lumbar spine, total hip, and femoral neck.”

·         What is truly missing in this manuscript is the inclusion of actual real-time data. Supplemental Table 3 is very descriptive; adding a Z score for the Lumbur spine, total hip, and femoral neck from individual patients to the table (time for DAXA approximate time when eGFR data collection will be ideal) would significantly enhance the manuscript.  
